# Association of Hospital Characteristics and Previous Hospitalization-Related Experiences with Patients’ Perceptions of Hospital Care in China

**DOI:** 10.3390/ijerph19137856

**Published:** 2022-06-27

**Authors:** Yufan Wang, Beizhu Ye, Yimei Zhu, Xiaoyu Wang, Yuan Liang

**Affiliations:** 1Department of Social Medicine and Health Management, School of Public Health, Tongji Medical College, Huazhong University of Science and Technology, Wuhan 430074, China; wyufan226@163.com (Y.W.); yebeizhu@163.com (B.Y.); wangxiaoyu361@163.com (X.W.); 2School of Media, Communication and Sociology, University of Leicester, Leicester LE1 7JA, UK; yz411@leicester.ac.uk

**Keywords:** patients’ perceptions of healthcare, healthcare quality, patient satisfaction, hospital management

## Abstract

Patients’ perceptions of healthcare vary over time and by setting, and previous studies have rarely focused on these factors. We aimed to measure patients’ perceptions of hospital care in China and to examine how patients’ perceptions of hospital care vary by hospital characteristics (differences in setting) and previous hospitalization-related experiences (changes with time). We conducted a national cross-sectional survey of 7267 inpatients between July 2014 and April 2015 in China. Hospital characteristics measured were hospital technical level, hospital type, teaching status, and the ratio of doctors/nurses to ward beds. Previous hospitalization-related experiences measured were current admission length, number of previous admissions, and hospital selection (hospital advertisements or personal recommendations). Patients’ perceptions of hospital care included perceptions of doctors, nurses, and hospital organization. Scores were highest for perceptions of nurses, followed by perceptions of doctors, and hospital organization. Of the five hospital characteristics rated, the technical level was most strongly associated with patient perceptions of healthcare. The effect of hospital admission length and frequency of hospitalization on patients’ perceptions was represented by a √-shaped dose–response curve (scores were initially high, then decreased, then rebounded to higher than the initial scores). Patients who selected a hospital with hospital advertisements gave lower scores than those without hospital advertisements, and patients who selected a hospital with personal recommendations gave higher scores than those without If the observed √-shaped dose–response curves indicate a causal relationship between patients’ perceptions and hospital admission length or frequency of hospitalization, this may help to guide the timing of patient satisfaction assessments. The negative association between patient perception and advertising, and the positive association with personal recommendations (word-of-mouth) and hospital technical level, could provide important information for clinicians and hospital administrators.

## 1. Introduction

Just as people need mirrors to clearly see themselves, healthcare providers require an external perspective on their services. As a measure of healthcare quality, patients’ perceptions of healthcare (commonly referred to as patient satisfaction) may provide such a perspective [1,2]. Patients’ perceptions of healthcare are a multi-dimensional construct encompassing numerous elements of healthcare, such as waiting time of pre-care, environment of wards, interactions with clinicians, availability of examination, and coordination of care. In addition, patients’ perceptions of healthcare, as a type of patient-reported outcome, are important not only in identifying physical and mental discomfort, but also in examining the process by which patients report symptoms of physical and mental discomfort to clinicians, including emerging or previously forgotten symptoms. Therefore, patients’ perceptions of healthcare reflect not only clinicians’ interpersonal communication skills, but also clinicians’ clinical interrogation ability [3,4]. With the advent of pay-per-performance and value-based reimbursement in the healthcare system, as well as the aging of the population and the chronicity of the disease, patients’ perceptions of healthcare are becoming increasingly important for healthcare providers and administrators [5,6].

Nonetheless, controversy exists regarding the interpretation and use of measures of patients’ perceptions of healthcare to improve the quality of healthcare and organizational management [7,8,9,10]. Differences in patients’ perceptions according to region and setting mean that most studies have examined predictors at the hospital level, such as nonprofit status, higher surgical volume, low mortality index, and low readmission rates (which often positively correlate with more positive patient perceptions) [4,7,11]. Previous research findings are inconsistent regarding the relationship between patient perceptions and some predictors, such as teaching status, number of beds, and the staff-to-patient ratio [12,13]. Patient-level predictors include age, sex, ethnicity, socioeconomic status, and chronic conditions, and existing studies often demonstrate divergent findings [5,6]. It is commonly accepted that patient perceptions of healthcare are impacted by their socioeconomic status, namely a lower education level with a more positive satisfaction rating. However, such frequently explored patient attributes are unsuitable as references or strategies to improve the quality of hospital care and patient satisfaction (usually as a risk adjustment) [13,14,15,16,17]. Furthermore, consumer perception is not static and is shaped both by sensory inputs from the current environment and by expectations generated from past experience [18,19]. Therefore, patients’ perceptions vary over time and are affected by factors such as previous hospitalization experiences and length of hospital stay. Previous studies have rarely focused on these factors. To our knowledge, only one survey of acute care general hospitals has investigated these aspects [20], and further empirical studies are needed on general care that use more rigorous methodologies to simultaneously incorporate hospital- and patient-level predictors to understand the complex nature of patients’ perceptions of healthcare [2,3,11].

To address the above gap, this study aimed to determine the association of hospital characteristics (differences in setting) and previous hospitalization-related experiences (changes with time) with patients’ perceptions of current hospital care, using data from a national sample of inpatients at general hospitals in China.

## 2. Methods

### 2.1. Study Design and Setting

We used stratified cluster sampling in 77 hospitals across seven provinces in China from July 2014 to April 2015. The details of this survey have been described in a previous report [21,22]. Briefly, we selected six provinces (Gansu, Yunnan, Jiangsu, Shandong, Hubei, and Guangdong) and Beijing, China’s capital, which have a combined population of 427.15 million, accounting for 31.88% of the total population of China. There were 85 eligible hospitals in the selected regions, of which 8 refused to participate, leaving a total of 77 participating hospitals (90.59%). In each hospital, convenience sampling was used to select patients from three to four surgical departments of different specialties and another three to four internal medicine departments, excluding obstetrics and pediatrics. A total of 528 departments were involved and the inpatients in the 528 departments were eligible to complete the survey. There were 24,250 eligible participants, of whom 11,884 did not complete the survey (49.01%). We excluded 4128 (17.02%) invalid questionnaires that contained errors or erratic responses. We also excluded 674 (2.78%) questionnaires with missing key variables. Following previous study [18], we also excluded 297 (1.22%) participants whose current admission length (days) was less than 1 day or more than 30 days. Finally, the analysis used data from the 7267 (29.97%) remaining responses (Appendix A). The three departments with the most participants were general surgery (1187), orthopedics (946), and cardiology (613) (Table 1). Participants provided oral informed consent for interviews. The institutional review board at the authors’ institutes approved the study protocol (No. IORG0003571).

### 2.2. Measures

Patients’ perceptions of hospital care were measured using a scale (Cronbach’s alpha, 0.844, Appendix A) comprising 11 items drawn mainly from the Consumer Assessment of Healthcare Providers and Systems (CAHPS) [8,9] and partly from the Picker Patient Experience Questionnaire [10]. The 11 items were categorized into three dimensions: care provided by doctors (4 items), by nurses (4 items), and by hospital organization (3 items: Clean environment, Quiet environment, and Convenience of medical exams). The doctor and nurse dimensions contained the same four items: (1) During your hospital stay, how often did doctors/nurses explain things in a way you could understand? (Communication); (2) When you had important questions to ask a doctor/nurse, was it difficult or easy to access your doctors/nurses? (Accessibility); (3) How much were you involved in medical/nursing services? (Involvement); (4) How much were doctors/nurses concerned about your mood? (Concern for patients’ mood). The service provided by the hospital organization was measured using three questions: (1) In general, during your hospital stay, how often was your ward cleaned? (Clean environment); (2) In general, during your hospital stay, how often was the area around your ward quiet? (Quiet environment); (3) How convenient were your medical exams during your current admission? (Convenience of medical exams). For each statement, patients were asked to indicate their perceptions on a 5-point scale (range of scores for each factor, 1–5; higher scores indicated more positive perceptions) (Appendix A). The patient experience domains were highly correlated overall (Cronbach’s alpha, 0.844); individual correlation coefficients ranged from 0.065 (correlation between communication with nurses and nurses’ concern for patients’ mood) to 0.647 (correlation between accessibility to doctors and doctors’ concern for patients’ mood) (Appendix A).

Although the CAHPS includes care provided by pharmacists [8,9], pharmacists in China have almost no direct contact with patients. As patients’ medication is prescribed by physicians, pharmacists are responsible only for the delivery of medication (according to physicians’ prescriptions) to the nurses’ ward station, so we did not examine their services in this study.

Previous hospitalization-related experiences were assessed using four statements: [22,23,24] (1) How many days have you been hospitalized in your current admission? (length of the current admission in days, with a blank filling question and divided into five levels: 1–3, 4–7, 8–14, 15–21, 22–30 days); (2) Excluding the current admission, how many times have you been hospitalized in this hospital in the last 3 years? (number of previous admissions (frequency), with five response options: 0, 1, 2, 3, and 4 and above); (3) Did you choose this hospital for your current admission because you saw an advertisement for it? (hospital advertisement: yes or no); (4) Did you choose this hospital for your current admission because it was recommended by relatives, friends, colleagues, etc.? (personal recommendation: called word-of-mouth in China: yes or no).

Hospital characteristics were assessed using five items: technical level of the current admission hospital (secondary or tertiary public hospital certified by the Chinese Ministry of Health (MOH)), hospital type (Western medicine (WM), or traditional Chinese medicine (TCM)), teaching status (teaching or non-teaching), and the ratio of doctors/nurses to ward beds. Data on hospital characteristics come from hospital official sources. Following the Chinese MOH and previous study [17], we divided ratios into three categories for doctors (≥0.3, 0.2–0.3, <0.2) and four for nurses (≥0.6, 0.5–0.6, 0.4–0.5, <0.4).

We collected most study data using patient questionnaires. Trained survey interviewers sent copies of the questionnaire to each department, with an explanation of the survey purpose and method. It was explained that participation was voluntary and that contributions would be anonymous. The survey was a self-administered paper survey, and family members were allowed to help patients fill in questionnaires. After 1 or 2 days, the survey interviewers returned to the department to collect completed questionnaires. We collected the data for the number of doctors, nurses, and beds in the department from the dean of the department or the nursing station. The data for hospital teaching status were obtained from the official website of hospitals and their affiliated universities. As some teaching hospitals in China are only nominal teaching hospitals (a status adopted to improve their social reputation), they were not included in the study. The status of teaching hospitals was verified by the administrative departments of hospitals and their affiliated universities.

Demographic characteristics measured included sex (male, female), age (15–29, 30–44, 45–59, ≥60 years), education level (middle school and below, high school, bachelor’s degree and above), marital status (married, unmarried, or other), medical insurance (yes, no), and self-reported economic status (good, fair, poor). A coding system was used to anonymously link patient data with the correct hospital and department.

### 2.3. Statistical Analysis

Data were weighted to adjust for nonresponses and improve the representativeness of the sample so that participants responding to the initial questions matched the demographic characteristics of the total hospitalization population issued by the National General Hospital in 2015 [25]. Owing to the limited parameters of the 2015 data reported by the National General Hospital, the present data were only weighted by age. To create condition-specific summary scores [9], we used a common method, in which the summary score is a percentage derived from individual actual scores from three dimensions, and the total (numerator) is divided, respectively, by the corresponding theoretical score from three dimensions and the total scores for patients’ perceptions of hospital care (denominator).

The key outcome variable in each analysis was patients’ perceptions of hospital care, comprising total scores, scores on the three dimensions, and scores on each of the 11 items. For crude comparisons, we used *t*-tests for continuous variables and chi-square tests for categorical variables. We then used a linear regression model and a binary logistic regression model to examine the association of hospital characteristics and patients’ previous hospitalization-related experiences with patients’ perceptions of hospital care (comprising total scores, dimension scores, and item scores). Negative parameter estimates in the model indicate lower (worse) adjusted mean patients’ perception percentiles, whereas positive estimates indicate higher (better) adjusted mean patients’ perception percentiles.

Although total scores and dimension scores for patients’ perceptions of hospital care are useful for general comparisons, and can provide operational guidance for reimbursement of Medicare rewards and penalties, these scores are too broad to provide detailed information for healthcare providers, especially clinicians. By contrast, an analysis of individual items can provide specific and targeted guidance for clinicians to evaluate their service behavior, so we further analyzed each of the 11 items in detail.

All models were adjusted for the following patient demographic characteristics, which have previously been associated with patients’ perceptions of healthcare: demographic [26,27,28,29,30]. sex, age, educational level, marital status, medical insurance, and self-reported economic status. Two-sided tests were used for all the analyses, and *p*-values of 0.05 or less were considered statistically significant. All analyses were performed using SPSS, version 22.0 (SPSS Inc., Chicago, IL, USA).

## 3. Results

The demographic characteristics of participants are summarized in Table 1. In the unweighted sample, the ratios of sex (male vs. female) and department (internal vs. surgical medicine) were almost half to half (50.23% vs. 45.86%, 50.05% vs. 49.95%, respectively). Patients aged 60 years and over accounted for about 33% of respondents; those with undergraduate education and above and who were unmarried/widowed accounted for about 20%, and those with medical insurance accounted for about 90%. The difference between the weighted and unweighted samples for almost all demographic characteristics except age and education level were not statistically significant; the weighted samples contained more participants over 60 years and more with an educational level of primary school and below (X^2^ = 109.372, *p* < 0.001; X^2^ = 12.803, *p* = 0.005, respectively). 

Overall, the highest scores (indicating more positive perceptions of hospital care) were for nurses (75.10%; 95%CI: 74.70% to 75.49%), followed by doctors (71.21%; 95%CI: 70.79% to 71.64%), and hospital organizational management (69.42%: 95%CI: 69.01% to 69.84%). The distribution of scores for all 11 items showed that the lowest (worst) ratings were for convenience of medical exams (17.90%), followed by clean environment (20.62%), and involvement in medical service (24.46%). For both the perception of doctor and nurse dimensions, the lowest ratings were for involvement in clinical care (24.46% for doctors and 27.88% for nurses); the highest rating for doctors was for communication (39.09%), followed by accessibility (30.88%); the highest rating for nurses was for accessibility (48.30%), followed by communication (45.96%) (Figure 1).

The analysis of total and dimension scores for patients’ perceptions by the five hospital characteristics indicated that the only statistically significant differences were for hospital technical level, with tertiary hospitals achieving higher scores (t = −4.128, *p* < 0.001; t = −4.631, *p* < 0.001; t = −3.310, *p* = 0.001; t = −4.706, *p* < 0.001, respectively). The differences by hospital type (WM vs. TCM) and doctor–bed ratio were not significant for any dimensions. The difference in hospital teaching status was statistically significant only for the doctor dimension, with non-teaching hospitals achieving higher scores (t = 2.991, *p* = 0.003). Additionally, scores on the nurse dimension (t = 4.063, *p* = 0.017) for the nurse–bed ratio ≥0.6 were significantly higher than scores for other nurse–bed ratio groups (Table 2).

We then examined each of the five hospital characteristics and the ratings on individual items in detail (Appendix A). Owing to space constraints, we only present the highest ratings for the five response levels of the 11 patient perception items by hospital characteristics and previous hospitalization-related experiences. Generally, the differences between ratings on the 11 items by the five hospital characteristics were similar to the scoring differences for the three dimensions and total score, and provide further insight into patient perceptions. The scoring differences between all 11 items by hospital levels were statistically significant; accessibility to doctors, quiet environment, and convenience of medical exams achieved higher ratings in tertiary hospitals. Of the 11 items, only accessibility to doctors, communication with nurses, and quiet environment were significantly correlated with teaching status; non-teaching hospitals had higher ratings. The scoring differences between the 11 items by hospital type were not statistically significant; however, clean environment and convenience of medical exams were rated lower in TCM hospitals. Similarly, the scoring differences between the 11 items by the doctor–bed ratio were not statistically significant; however, a quieter environment was associated with a doctor–bed ratio ≥ 0.3, and greater convenience of medical exams was associated with a doctor–bed ratio < 0.2. The nurse–bed ratio was not significant for all items analyzed.

Multivariate linear regression analysis showed significant differences between doctor dimension scores, nurse dimension scores, and total scores by hospital level (*ß* = 1.53, 95%CI: 0.1 to 2.96; *p* = 0.036; *ß* = 2.01, 95%CI: 0.65 to 3.38; *p* = 0.004; *ß* = 1.46, 95%CI: 0.28 to 2.64; *p* = 0.015, respectively) (Table 3). Consistent with the results of the univariate analysis, there was no significant difference between the three dimensions and total score by hospital type. There were significant score differences by teaching status only for the doctor dimension (*ß* = −2.36, 95%CI:−3.48 to −1.24; *p* < 0.001). The difference for the doctor dimension and the total score remained significant in the multivariate analysis (*ß* = −2.34, 95%CI: −3.76 to −0.92; *p* = 0.001; *ß* = −1.20, 95%CI: −2.37 to −0.04; *p* = 0.044, respectively). Contrary to patients whose hospital selection was not based on personal recommendation, those whose selection was based on personal recommendation reported higher scores (ß _total scores_ = 1.02, 95%CI: 0.27 to 1.77; *p* = 0.008).

The data in Table 2 show that, compared with patients whose hospital selection for the current admission was based on hospital advertisements, those whose selection was not based on hospital advertisements gave higher (better) scores on the doctor dimension, the nurse dimension, and total scores (t = 4.035, *p* < 0.001; t = 2.561, *p* =0.011; t = 2.283, *p* = 0.023, respectively). Table 2 also shows that patients who had previous hospitalization experience provided lower ratings than those without previous hospitalization experience, which is consistent with previous findings [13]. The mean length of the current hospital admission was 8.35 ± 6.16 days, and the median was 7 days. Figure 2 shows the dose–response relationship between the care score reported by patients by length of hospital admission and number of previous hospitalizations. For length of hospital admission, in the first 3 days, the scores were relatively high. However, on the 4th to 7th days, there was a downward trend in scores. One week later, scores continued to increase and exceeded the score of the first 3 days, showing a √-shaped (not U-shaped) dose–response curve. The score differences on the doctor dimension and the total score were statistically significant. For the number of previous hospitalizations, the √-shaped dose–response curve was more obvious, and the score differences for the doctor dimension, nurse dimension, hospital organization dimension, and the total score were statistically significant. In the multivariate analysis, this pattern of a drop in scores followed by an increase was generally consistent, and score differences on the doctor dimension by length of hospital admission and number of previous hospitalizations were statistically significant (Table 3).

As described above, we then examined ratings on the four items assessing previous hospitalization-related experience and patients’ perception ratings in detail (Appendix A). Compared with patients whose hospital selection was based on hospital advertisements, those whose selection was not based on advertisements reported higher scores for communication and accessibility (X^2^ = 88.089, *p* < 0.001; X^2^ = 25.992, *p* < 0.001, respectively), but lower scores for concern and involvement (X^2^ = 26.405, *p* < 0.001; X^2^ = 14.352, *p* = 0.006, respectively) on the nurse dimension. These individuals gave similar scores on the doctor dimension (except for the concern item). Compared with patients whose hospital selection was based on hospital advertisements, those who had not selected based on advertisements reported lower scores on clean environment, quiet environment, and convenience of medical exams (the differences were significant for clean environment and quiet environment: X^2^ = 24.499, *p* < 0.001; X^2^ = 9.785, *p* = 0.044, respectively). Items that remained significant in the multivariate analysis were communication with doctors and nurses (OR = 0.75, 95%CI: 0.64 to 0.89; *p* = 0.001; OR = 0.78, 95%CI:0.67 to 0.92; *p* = 0.003, respectively: lower ratings for patients whose selection was based on advertisements), nurses’ concern, and involvement in nursing services (OR = 1.30, 95%CI: 1.10 to 1.54; *p* = 0.002; OR = 1.22, 95%CI:1.03 to 1.45; *p* = 0.022, respectively: higher ratings for patients whose selection was based on advertisements). Patients whose selection was based on personal recommendation reported higher scores on all 11 items. Most of these differences were significant, except communication with doctors, accessibility to nurses, and convenience of medical exams. After adjustment in the multivariate analysis (Appendix A), only the items doctors’ concern for patients’ mood, nurses’ concern for patients’ mood (OR = 1.21, 95%CI: 1.07 to 1.36; *p* = 0.002; OR = 1.16, 95%CI:1.04 to 1.30; *p* = 0.009, respectively), and involvement in nursing services (OR = 1.14, 95%CI: 1.02 to 1.28; *p* = 0.024) remained significant.

Differences on the 11 items by length of hospital admission were generally consistent with differences on the three dimensions and total score; that is, the scoring pattern was initially high, then decreased, then rebounded to a higher level than the initial scores. Additionally, doctors’ and nurses’ concern (OR_≥22days_ = 1.60, 95%CI: 1.22 to 2.10; *p* < 0.001; OR = 1.35, 95%CI:1.03 to 1.76; *p* = 0.028; respectively) and involvement (OR_≥22days_ = 1.50, 95%CI: 1.13 to 1.98; *p* = 0.005; OR = 1.50, 95%CI:1.14 to 1.96; *p* = 0.003, respectively), and convenience of medical exams (OR_≥22days_ = 1.36, 95%CI: 1.10 to 1.85; *p* = 0.047) showed significant differences. Similarly, differences on the 11 items by number of previous hospitalizations were generally consistent with those on the three dimensions and total score (i.e., the scoring pattern was initially high, then decreased, then rebounded to a higher level than the initial scores, and doctors’ and nurses’ accessibility (OR_≥4times_ = 1.29, 95%CI: 1.05 to 1.59; *p* = 0.014; OR = 1.25, 95%CI: 1.03 to 1.52; *p* = 0.027, respectively), concern (OR_≥4times_ = 1.38, 95%CI: 1.12 to 1.71; *p* = 0.002; OR = 1.32, 95%CI: 1.07 to 1.62; *p* = 0.009, respectively), and involvement (OR_≥4times_ = 1.51, 95%CI: 1.22 to 1.87; *p* < 0.001; OR = 1.42, 95%CI: 1.15 to 1.75; *p* = 0.001, respectively) showed significant differences (Appendix A).

We also found significant cross-regional differences in patients’ perceptions of hospital care (Appendix A). Regions in eastern China had the highest scores (total score 74.3%, 95% CI: 73.44% to 74.95%) and regions in central China had the lowest scores (total score 68.96%, 95% CI: 68.13% to 69.79%). There was a similar range in the percentage of patients who rated their care highly. For example, for doctors’ concern for patients’ mood, there was a more than 10 percentage point difference between the best (33.24%) and worst (20.70%) regions.

## 4. Discussion

To our knowledge, this study provides the first national data on patients’ perceptions of healthcare in China. Overall, the highest (best) ratings of healthcare were found for nurse care, and the lowest (worst) ratings for hospital organization. For care by both doctors and nurses, the lowest ratings were for involvement in care (24.46% and 27.88%, respectively). Scores of patient perceptions of hospital care by length of hospital admission and number of previous hospitalizations showed a √-shaped dose–response curve. Of the five hospital characteristics rated, the technical level was most strongly associated with patient perceptions of healthcare. The association of patient perceptions with advertising was negative and that with personal recommendations was positive.

Patients’ ratings of hospital care in China were relatively low compared with similar ratings in previous studies, particularly ratings in the US report [7,9,27,31]. There are two main possible explanations. First, as market-oriented reform of medical services in China began in 1985, most hospitals, including public hospitals, need to increase revenue and reduce expenditure. Therefore, hospitals often need to try to reduce the number of medical staff or persuade them to work full-time. Sometimes medical staff are overloaded and doctors in particular often must see as many patients as possible. Although the market-oriented reform of medical services stopped in 2005 [32], most hospitals need to generate their own income owing to lack of government investment. The so-called “three long, one short” conditions (i.e., long registration and queue times; long waiting times; long dispensary and payment queue times; and short physician visit times) and the corresponding deterioration in patient–provider relationships in China are testimony to the low opinion of healthcare by patients in China [22,33]. Second, unlike in the US, patients in China are completely free to visit hospitals, including outpatient clinics, and go directly to the hospital without seeing a primary care provider in the community. This not only increases the burden on hospital doctors, but also aggravates their job burnout, which may lead to a lower level of patient-oriented care of doctors in China than that in the US, especially in hospital settings. Third, the current practice in the US of publicly reporting data on the quality of healthcare, and incentivizing (or penalizing) clinicians based on patient satisfaction data, act as external motivations to improve the quality of healthcare, but these practices are not followed in China. Additionally, patient-centered care in the US is advanced compared with similar care in China; however, lessons are being learned from the US, and healthcare in China is slowly developing. In the three dimensions of patients’ perceptions of hospital care, the service provided by hospital organizations is the lowest score; furthermore, the convenience of medical exams is the lowest score among the three items of the hospital organization dimension. This may further verify the above-mentioned “three long, one short”, and more importantly, it also reveals the shortcomings of the service provided by the hospital organization dimension, compared to those by the doctor and nurse dimension [22,33]. Among the four items of the doctor/nurse dimension, the scores of nurses are higher than those of doctors, especially the accessibility, which may be related to a higher nurse-to-bed ratio and more emphasis on patient-oriented services by nursing policies.

The demographic characteristics of patients’ perceptions with significant differences are consistent with most previous studies [5,6]. For example, patients with poor self-reported economic status reported lower perceptions of hospital care, which may be due to their physical distress combined with their psychosocial distress, and those may be the key target group of Medicaid policy.

The technical level of each hospital in China is certified by the government, and mainly comprises the quality of clinical care (i.e., the professional competence of clinicians, rather than medical equipment) and hospital organization (although clinical care quality is dominant) [20,22]. Information about hospital technical levels is in the public domain. The positive association between the hospital technical level and patients’ perceptions of hospital care in this study may indicate a positive association between the quality of clinical care and patient reporting quality, which is consistent with previous study findings from different countries and regions [3,4]. As described above, patients’ perceptions of healthcare reflect not only clinicians’ interpersonal communication skills, but also on clinicians’ clinical interrogation skills. Therefore, the quality of patient reports not only improves patient satisfaction and rewards for medical insurance (an external mechanism for quality improvement), but also informs the development of clinical interrogation and professional competence (an internal mechanism for quality improvement). Clinicians should be more concerned with internal mechanisms than with external mechanisms, as internal mechanisms are more relevant to the task of improving clinical quality.

Although TCM has been practiced for several thousand years in China, and the new Chinese National Health Guiding Principles emphasize the importance of both TCM and WM [34]; the development of WM has been rapid in the past decades; in contrast, the development of TCM has been very slow (even retrogressive) in some areas. TCM practice is currently learning from WM, and to a certain extent, even emulating WM practice. These factors may explain why we found no significant difference in patients’ perceptions of TCM and WM healthcare. It is worrying that the current combination of TCM and WM seems to be evolving into an assimilation of TCM into WM. Understanding and developing the traditional wisdom of TCM to serve human health is the responsibility and mission of healthcare providers and managers, and TCM may play a special role in addressing the suffering of patients in the future.

Compared with predictors of hospital characteristics, predictors of previous hospitalization-related experiences had more impact on patient ratings, particularly length of hospital admission and number of previous hospitalizations. Patients’ perceptions of healthcare related to these two factors showed a √-shaped dose–response curve, and the association largely remained significant after adjustment for other factors. The dose–response curves may indicate changes in consumer perceptions over time. The initial high score, regardless of whether it reflects the first hospitalization occurrence or the first 1–3 days of hospitalization, may be related to the novelty of the situation and a corresponding positive attitude in patients. Subsequently, this attitude may change; the patient may not experience immediate relief for their illness (they may be undergoing tests and awaiting the results of exams, but not receiving treatment) and may experience a sense of loss and anxiety, leading to a reduction in subsequent scores. As time passes and their treatment progresses, the patient may experience an improvement in their condition and develop a deeper understanding and trust in the clinicians, leading to higher scores. Specifically, regarding the length of hospital admission, patients’ scores within 7 days of admission may be lowest; regarding the number of previous hospitalizations, the lowest scores may be from patients who have had one hospitalization. The average length of hospital stay in 2014 in China was 9.6 days [35]. Therefore, the above factors suggest that it may be more reasonable to measure the perception of healthcare in patients who have been hospitalized for more than 7 days or who have had one hospitalization experience, indicating the importance of timing in assessing patients’ perceptions of healthcare, which may be worthy of further study in the future.

Patients in China are free to choose which doctor to consult. Therefore, many hospitals, including public hospitals, advertise for patients. Interestingly, only 11.5% of patients in this study selected the hospital for their current hospitalization based on advertisements, and 45.5% of patients selected their hospital based on personal recommendation (word-of-mouth). More interestingly, there was a negative association between patient perceptions and advertising, and a positive association between patient perceptions and personal recommendations. A possible explanation is that the advertisements issued by hospitals tend to highlight only the positive qualities of the hospital. This may mean that patients’ expectations tend to be high before admission; after admission, they are more aware of both the positive and negative qualities of the hospital, so their ratings decrease. In contrast, personal recommendations (unlike advertising) may be more balanced, and may include both positive and negative qualities.

A low doctor–bed ratio or nurse–bed ratio avoids overstaffing but increases the workload (and may even lead to overload) of doctors and nurses in China [21]. We generally found no significant differences in patients’ perceptions of healthcare according to the doctor–bed ratio or nurse–bed ratio. A possible explanation may be related to income allocation. As mentioned above, hospitals need to increase revenue and reduce expenditure; the more staff in a department, the less income can be allocated. Conversely, the fewer staff, the more income can be allocated. Therefore, the size of the doctor/nurse–bed ratio may not impact perceptions of healthcare quality.

## 5. Strengths and Limitations

This study had a number of strengths and limitations. First, our study extends this line of inquiry by focusing on a range of topics, especially hospital characteristics (differences in setting) and previous hospitalization-related experiences (changes with time). Second, to our knowledge, this study provides the first national data on patients’ perceptions of healthcare in China. Third, private hospitals were not included in the study. Private hospitals in China are mainly specialized hospitals, such as beauty hospitals, eye hospitals, etc. The hospitalization services provided by private hospitals accounted for only 12.65% of the total inpatient services in China [26]. Fourth, the social network and information channel are not covered in this survey, and are the direction of future research [36,37]. Fifth, our study relied on cross-sectional data and therefore causality cannot be established.

## 6. Conclusions

Using a nationally representative sample, this study extends previous research that has described disparities in patients’ perceptions of healthcare for both hospital characteristics and previous hospitalization-related experiences. The results suggest that the technical level of the hospital is the factor most strongly associated with patients’ perceptions of healthcare, rather than any hospital characteristics, which indicates that the quality of patient reports is important not only to improve patient satisfaction and medical insurance reimbursement, but also to increase the quality of clinicians’ professional competence. 

Patients’ perceptions of healthcare vary not only according to settings, but also across time, reflecting a √-shaped dose–response curve of patients’ perceptions by length of hospital admission and number of previous hospitalizations. If it could be established that these associations were causal, this would help in determining the timing of patient satisfaction assessments. Additionally, the negative correlation between patient perception and advertising, and the positive correlation between patient perception and personal recommendations (word-of-mouth), could also provide important information for clinicians and hospital administrators.

## Figures and Tables

**Figure 1 ijerph-19-07856-f001:**
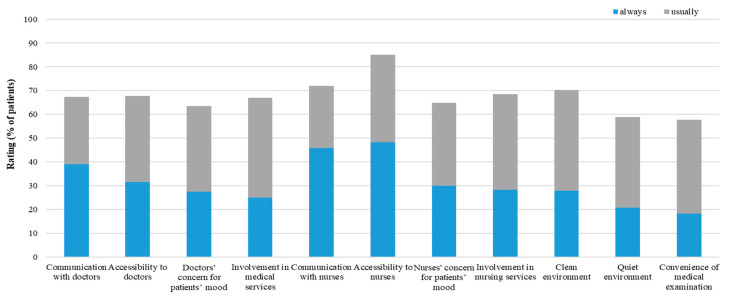
Percentage of patients’ perceptions of hospital care.

**Figure 2 ijerph-19-07856-f002:**
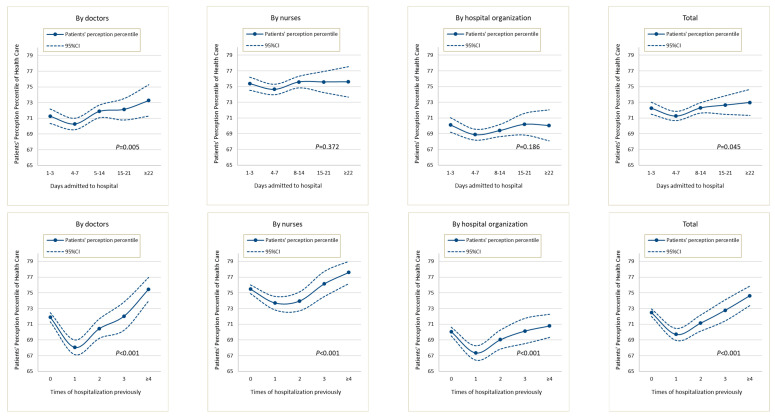
Dose–response curves of patients’ perceptions with days admitted to hospital and times of hospitalization previously.

**Table 1 ijerph-19-07856-t001:** Socio-demographic characteristics of participants.

	Unweighted Samples	Weighted Samples	X^2^	*p*
Gender				
Male	3650 (50.23)	3653 (50.26)	0.003	0.959
Female	3333 (45.86)	3330 (45.83)		
missing	284 (3.91)	284 (3.91)		
Age (years old)				
15–44	2782 (38.28)	2408 (33.14)	109.372	<0.001
45–59	2075 (28.55)	1840 (25.32)		
≥60	2410 (33.16)	3019 (41.54)		
Education status				
Primary school and lower	1440 (19.82)	1602 (22.05)	12.803	0.005
Junior high school	2083 (28.66)	2083 (28.66)		
Senior high school	2123 (29.21)	2045 (28.14)		
Undergraduate and above	1535 (21.12)	1445 (19.88)		
missing	86 (1.18)	92 (1.27)		
Marital status				
Married	5793 (79.72)	5796 (79.76)	0.009	0.923
Single/divorced/widowed/other	1411 (19.42)	1406 (19.35)		
missing	63 (0.87)	65 (0.89)		
Self-reported economic status				
Good	1958 (26.94)	1988 (27.36)	0.984	0.611
Fair	3974 (54.69)	3986 (54.85)		
Bad	1274 (17.53)	1231 (16.94)		
missing	61 (0.84)	62 (0.85)		
Medical insurance				
Yes	6440 (88.62)	6464 (88.95)	0.634	0.426
No	631 (8.68)	604 (8.31)		
missing	196 (2.70)	199 (2.73)		
Special				
Internal medicine	3637 (50.05)	3737 (51.42)	2.753	0.097
Neurology	302 (4.16)	316 (4.35)		
Respiratory medicine	410 (5.64)	438 (6.03)		
Cardiology	613 (8.44)	672 (9.25)		
Gastroenterology	598 (8.23)	600 (8.26)		
Endocrinology	442 (6.08)	437 (6.01)		
Nephrology	223 (3.07)	225 (3.10)		
Hematology	129 (1.78)	123 (1.69)		
Oncology	141 (1.94)	142 (1.96)		
Rheumatology	115 (1.58)	109 (1.50)		
General internal medicine	195 (2.68)	201 (2.77)		
Others	469 (6.45)	472 (6.50)		
Surgery	3630 (49.95)	3530 (48.58)		
Brain surgery	204 (2.81)	199 (2.74)		
Thoracic/cardiac surgery	311 (4.28)	318 (4.37)		
General Surgery	1187 (16.31)	1168 (16.07)		
Urology	360 (4.95)	362 (4.99)		
Orthopedics	946 (13.02)	914 (12.58)		
Gynecology	278 (3.83)	249 (3.42)		
Otolaryngology	138 (1.90)	128 (1.77)		
Others	206 (2.83)	192 (2.64)		

**Table 2 ijerph-19-07856-t002:** Patients’ perceptions of hospital care according to hospital characteristics and patients’ previous hospitalization-related experiences.

	%	Provided by Doctors		Provided by Nurses		Provided by Hospital Organization		Total	
		Mean Score (95%CI)	*p* Value	Mean Score (95%CI)	*p* Value	Mean Score (95%CI)	*p* Value	Mean Score (95%CI)	*p* Value
Total		71.21 (70.79,71.64)		75.1 (74.7,75.49)		69.42 (69.01,69.84)		71.91 (71.56,72.26)	
**Hospital characteristics**									
Hospital technical level									
SH	11.76	68.84 (67.64, 70.03)	<0.001	72.53 (71.35, 73.71)	<0.001	67.51 (66.27, 68.75)	0.001	69.63 (68.63, 70.63)	<0.001
TH	88.24	71.53 (71.08, 71.98)		75.44 (75.02, 75.86)		69.68 (69.24, 70.12)		72.22 (71.85, 72.59)	
Hospital type									
WM	71.98	71.05 (70.55, 71.56)	0.238	75.10 (74.63, 75.57)	0.995	69.49 (69.01, 69.98)	0.593	71.88 (71.47, 72.30)	0.790
TCM	28.02	71.62 (70.84, 72.41)		75.10 (74.36, 75.83)		69.24 (68.46, 70.02)		71.99 (71.34, 72.63)	
Teaching status									
Non-teaching	77.99	71.56 (71.08, 72.04)	0.003	74.93 (74.48, 75.38)	0.125	69.34 (68.87, 69.81)	0.462	71.94 (71.55, 72.34)	0.731
Teaching	22.01	69.99 (69.07, 70.92)		75.68 (74.84, 76.52)		69.72 (68.84, 70.59)		71.80 (71.05, 72.54)	
Ratio of doctors to ward beds									
<0.20	32.86	71.43 (70.69, 72.17)	0.764			68.98 (68.24, 69.71)	0.187	70.2 (69.56, 70.84)	0.687
0.20–0.30	42.56	71.06 (70.40, 71.72)				69.43 (68.81, 70.06)		70.25 (69.69, 70.8)	
≥0.30	24.58	71.19 (70.35, 72.04)				70.01 (69.17, 70.85)		70.6 (69.86, 71.34)	
Ratio of nurses to ward beds									
<0.4	59.36			75.22 (74.70, 75.73)	0.017	69.63 (69.09, 70.17)	0.234	72.42 (71.96, 72.88)	0.396
0.4–0.6	34.76			74.56 (73.89, 75.24)		69.29 (68.6, 69.99)		71.93 (71.33, 72.53)	
≥0.6	5.88			77.06 (75.48, 78.63)		68.13 (66.45, 69.81)		72.59 (71.17, 74.01)	
**Previous hospitalization-related experiences**									
Number of previous admissions in the last three years									
No (0)	53.13	71.89 (71.31, 72.47)	0.001	75.47 (74.92, 76.02)	0.058	70.06 (69.49, 70.62)	0.001	72.47 (71.99, 72.95)	0.001
Yes (≥1)	45.41	70.37 (69.75, 71.00)		74.69 (74.11, 75.28)		68.68 (68.07, 69.3)		71.25 (70.74, 71.76)	
Hospital selection by personal recommendations									
No	53.13	71.02 (70.44, 71.60)	0.357	74.87 (74.33, 75.42)	0.133	68.91 (68.34, 69.47)	0.006	71.60 (71.13, 72.07)	0.041
Yes	45.36	71.43 (70.79, 72.06)		75.49 (74.90, 76.08)		70.09 (69.47, 70.7)		72.33 (71.81, 72.85)	
Hospital selection by advertisements									
No	86.16	71.47 (71.01, 71.92)	<0.001	75.34 (74.91, 75.76)	0.011	69.36 (68.92, 69.8)	0.566	72.05 (71.68, 72.42)	0.023
Yes	11.62	68.73 (67.43, 70.04)		73.63 (72.39, 74.87)		69.76 (68.45, 71.07)		70.71 (69.61, 71.80)	

Note: WM, Western Medicine. TCM, Traditional Chinese Medicine. TH, Tertiary Hospital. SH, Secondary Hospital.

**Table 3 ijerph-19-07856-t003:** Multivariate linear regression model to examine the association of hospital characteristics and patients’ previous hospitalization-related experiences with patients’ perceptions of hospital care.

	Provided by Doctors		Provided by Nurses		Provided by Hospital Organization		Total	
	*β* (95%CI)	*p* Value	*β* (95%CI)	*p* Value	*β* (95%CI)	*p* Value	*β* (95%CI)	*p* Value
**Hospital characteristics**								
Hospital technical level (ref = Secondary hospital)								
Tertiary hospital	1.53 (0.10, 2.96)	0.036	2.01 (0.65, 3.38)	0.004	1.02 (−0.4, 2.44)	0.158	1.46 (0.28, 2.64)	0.015
Hospital type (ref= WM)								
TCM	0.44 (−0.58, 1.46)	0.396	−0.59 (−1.56, 0.38)	0.231	−0.99 (−2, 0.02)	0.054	−0.41 (−1.24, 0.43)	0.344
Teaching status (ref = Non-teaching)								
Teaching	−2.36 (−3.48, −1.24)	<0.001	0.38 (−0.70, 1.46)	0.491	0.17 (−0.95, 1.29)	0.768	−0.55 (−1.49, 0.38)	0.246
Ratio of doctors to ward beds (ref = <0.20)								
0.20–0.30	−0.48 (−1.53, 0.57)	0.369			0.44 (−0.61, 1.49)	0.415	0.17 (−0.70, 1.04)	0.704
≥0.30	0.12 (−1.08, 1.32)	0.844			1.52 (0.25, 2.78)	0.019	0.97 (−0.09, 2.02)	0.073
Ratio of nurses to ward beds (ref = <0.4)								
0.4–0.6			−0.57 (−1.49, 0.35)	0.225	−0.73 (−1.72, 0.27)	0.151	−0.92 (−1.75, −0.09)	0.029
≥0.6			1.95 (0.09, 3.80)	0.040	−2.2 (−4.22, −0.17)	0.033	−0.27 (−1.95, 1.42)	0.756
**Previous hospitalization-related experiences**								
Current admission length (days) (ref = 1–3 days)								
4–7	−0.85 (−2.08, 0.38)	0.174	−0.59 (−1.75, 0.57)	0.317	−1.02 (−2.23, 0.18)	0.096	−0.82 (−1.83, 0.18)	0.108
8–14	0.39 (−0.89, 1.67)	0.548	0.33 (−0.88, 1.55)	0.590	−0.62 (−1.88, 0.64)	0.334	0.04 (−1.01, 1.09)	0.936
15–21	1.09 (−0.62, 2.81)	0.212	0.20 (−1.42, 1.83)	0.805	−0.09 (−1.78, 1.60)	0.917	0.39 (−1.02, 1.80)	0.590
≥22 days	2.73 (0.49, 4.97)	0.017	1.28 (−0.84, 3.40)	0.237	0.66 (−1.55, 2.86)	0.559	1.56 (−0.27, 3.40)	0.095
Number of previous admissions in the last three years (ref = 0)								
1	−4.07 (−5.23, −2.91)	<0.001	−1.54 (−2.64, −0.45)	0.006	−3.07 (−4.21, −1.93)	<0.001	−2.89 (−3.84, −1.94)	<0.001
2	−1.71 (−3.17, −0.24)	0.022	−1.17 (−2.55, 0.22)	0.099	−1.25 (−2.70, 0.19)	0.088	−1.38 (−2.58, −0.17)	0.025
3	−1.27 (−3.34, 0.80)	0.230	0.55 (−1.40, 2.51)	0.580	−0.94 (−2.98, 1.1)	0.366	−0.53 (−2.23, 1.16)	0.538
≥4 times	2.67 (0.91, 4.43)	0.003	2.32 (0.65, 3.98)	0.007	0.4 (−1.34, 2.14)	0.650	1.78 (0.33, 3.22)	0.016
Hospital selection by personal recommendations (ref = No)								
Yes	1.08 (0.16, 1.99)	0.021	0.82 (−0.04, 1.69)	0.063	1.20 (0.30, 2.10)	0.009	1.02 (0.27, 1.77)	0.008
Hospital selection by advertisements (ref= No)								
Yes	−2.34 (−3.76, −0.92)	0.001	−1.34 (−2.68, 0.01)	0.051	0.12 (−1.28, 1.52)	0.863	−1.20 (−2.37, −0.04)	0.044

Note: All models were adjusted for the following patient demographic characteristics: sex, age, educational level, marital status, medical insurance, and self-reported economic status.

## Data Availability

The study database is available via e-mail to the corresponding authors: Yuan Liang [liangyuan217@hust.edu.cn].

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
