# Peer review of "Association of Hospital Characteristics and Previous Hospitalization-Related Experiences with Patients’ Perceptions of Hospital Care in China"

_ijerph, 2022, doi:10.3390/ijerph19137856_

Round 1

Reviewer 1 Report

I would like to thank the editor of this prestigious journal for the opportunity to evaluate this study and I would also like to congratulate the researchers for the effort made and the results obtained.

First of all, I think the study is interesting and may open the door to designing new strategies to address these information gaps by health care institutions. 

The introduction does not address the importance and weight of social networks. Nowadays, healthcare institutions have social networks and they are an excellent channel for communicating with users and finding out their preferences and opinions of the service received. Moreover, its correct use limits the fraudulent use made by healthcare users, taking into account the comments made by social network users on health-related issues more than the institutions themselves.

 Harris, J.; Atkinson, A.; Mink, M.; Porcellato, L. Young People's experiences and perceptions of YouTuber-produced health content: Implications for health promotion. Health Educ. Behav. 2020, 48, 199-207.
Social networking

Sampson, M.; Cumber, J.; Li, C.; Pound, C.M.; Fuller, A.; Harrison, D. A systematic review of methods for studying consumer health YouTube videos, with implications for systematic reviews. PeerJ 2013, 1, e147. [CrossRef].

If there are differences in social media use in China compared to other countries, now is a good time to explain it.

The study population is very good and the methodology has been rigorously conducted. Given the magnitude of the study, I miss the fact that the respondents were not asked about their social network use in China. 

In the limitations section of the study, I would point out that the channels through which patients receive information have not been analysed and this could be a study to be carried out in the future, to improve the communication systems in social networks that hospitals use with their users and to be able to build loyalty and improve the opinion of users with respect to the service.

I congratulate the researchers for the excellent research they have carried out and I am grateful for the opportunity to have been able to evaluate this study.

Author Response

please see 

Reviewer 2 Report

The article offers an original and interesting analysis of patients’ perception of (public) hospital care in China, and certainly deserves to be published.
The analysis is scientifically sound and methodologically correct.
In this sense, my opinion is intermediate between accept in its present form and accept after minor revisions.

In order to further improve the quality of the article, I have suggested some additions.

More specifically, in the Discussion, the authors refer to the 1985 marked-oriented reform of medical services in China. It would therefore be useful to know (especially in relation to the quality of the health service) why this reform was introduced and why it then stopped in 2005.

In the Discussion, a more extensive analysis of the reported “two main possible explanations” (see page 10, lines 14-27)  would be useful. 

Considering that patients’ perception of healthcare can change by regions, some information on the territorial distribution of the hospitals included in the sample should be added. 

Author Response

please see the word file uploaded.

Reviewer 3 Report

This article entlited "Association of Hospital Characteristics and Previous Hospitalization-related Experiences with Patients’ Perception of Hospital Care in China" aimed to measure patients’ perception of hospital care in China and to examine how patients’ perception of hospital care vary by hospital characteristics(differences in setting) and previous hospitalization-related experiences(changes with time).

This article is very interesting and well conducted.

Please, improve the issues below:

  • Replace the S Fig.  by a figure with better resolution.
  • The same for Figure 1 and 2.
  • Improve Introduction rationale.  What are the studies already done about this topic?
  • Measures (2.2) description should be improved.

Author Response

please see the word file uploaded

Reviewer 4 Report

This study provides an explanation from various attribute information in terms of how inpatients' evaluation of Chinese hospitals is influenced. In particular, the experience of hospitalization was shown to shift inpatient evaluations along a root-shaped curve. The findings are presented that contribute to the handling of inpatients in the context of their situation in the long-term service of hospitalization.

However, from the perspective of an academic paper, there are scattered points that need to be corrected.

As the main missing part, the author(s) do not provide a detailed description of the research issues, or research gap, to be addressed in this study. The author (s) should include a section related to a literature review and provide a detailed description of related research works to date and the issues to be addressed in this study. 

They also do not provide details of the research design, such as how the analysis addressed in this study is intended to be set up and why these analysis methods are appropriate. The methodology section should also explain the overall analysis used in this study and its validity.

Also, the discussion section does not clearly explain where the academic novelty of this study lies. The author(s) should describe the newness of the findings so that the reader can clearly understand where the novelty lies. In addition, the theoretical and practical implications based on such findings of this study should also be clearly discussed.

Below are detailed comments tied to the page number and line number.

---------------------

# Page 1, line 11: Abstract
The abstract should explicitly indicate the research issue, or research gap, that the author(s) are addressing to solve in this study.

# Page 1, line 13: Abstract
"... 2015in China." should be "... 2015 in China.". 

# Page 1, line 18: Abstract
In the abstract, specific figures in the statistical analysis can be omitted. This is because the main body has them. 

# Page 1, line 29: Abstract
The author(s) argues "this may help to guide the timing of patient satisfaction assessments" as an implication. But is this implication fair? Are author(s) suggesting that they refrain from evaluating in situations where the evaluation could be considered low? If so, it cannot be supported from an academic standpoint.

# Page 2, line 58: 1. Introduction/Par.1: 
The author(s) claim that the factors such as previous hospitalization experiences and length of hospital stay, is rarely addressed in the extant literature. However, this paper does not have a section describing existing research efforts. Thus, in a section such as a literature review, the author(s) should reiterate the existence of such a research gap after detailing the efforts of existing studies in related fields.

# Page 2, line 68: 2.1. Study Design and Setting/Par.1: 
The data presented here is from seven years ago and may differ from the current situation. The most recent relevant studies should also be discussed in the literature review section so that the data of this survey is still valid.

# Page 2, line 73: 2.1. Study Design and Setting/Par.1: 
Since the meaning of "S Fig" is not clear at first glance, a description should also be added to indicate that it is a supplement.

# Page 3, line 82: 2.2. Measures/Par.1: 
"... mood.." should be "... mood.". 

# Page 3, line 91: 2.2. Measures/Par.3: 
What is the "academic status" here? A more detailed explanation should be added so that the reader can take in the meaning.

# Page 3, line 95: 2.2. Measures/Par.4: 
Is the term "sociodemographic" common use? It simply makes sense with demographics.

# Page 3, line 98: 2.3. Statistical Analysis/Par.1
I did not understand the meaning of "Data were weighted to adjust for nonresponses". Could the author(s) please explain more clearly the intent of this statement?

# Page 4, line 110: 3. Results/Par.1:

If the author(s) present the results of a statistical test, please specify what statistical test they used. This is because the derivation of the p-value is different for each statistical analysis. 

# Page 4, line 113: Table 1. 
Although the chi-square values are shown in the table, what is shown in the table should also be mentioned in the main text. 

# Page 5, line 123: 3. Results/Par.3: 
Please also specify the analytical methods used here and why the author(s) are using these methods here. But this should be explained in the Methodology section.

# Page 5, line 129: Table 2. 

It is not obvious at first glance how this table is structured. Please revise it so that the reader can immediately read the meaning of this table. 

# Page 7, line 132: 3. Results/Par.4: 

Again, please also specify the analytical methods used here and why the author(s) are using these methods here. But this should be explained in the Methodology section.

# Page 7, line 141: Table 3. 
It is not obvious at first glance how this table is structured. If it is the result of multiple regression analysis, the results should be presented in a common format.

# Page 9: Figure 2. 
A part of a figure is missing. Please correct them so that they are not unclear to the reader.

# Page 10, line 8: 4. Discussion
In discussion, please do not just assert the results of the current analysis, but describe them in a way that clearly identifies what are the novel findings.

# Page 10, line 8: 4. Discussion

The discussion section should also discuss contributions to existing research based on the novel results obtained in this study. This is called theoretical implication. 

# Page 10, line 8: 4. Discussion

The discussion section should also discuss contributions to practice based on the novel results obtained in this study. This is called practical/managerial implication. 

---------------------

Author Response

please see the word file uploaded

Round 2

Reviewer 3 Report

All my comments were addressed by the authors. Congrats!